# Physiotherapy Programmes Aided by VR Solutions Applied to the Seniors Affected by Functional Capacity Impairment: Randomised Controlled Trial

**DOI:** 10.3390/ijerph19106018

**Published:** 2022-05-15

**Authors:** Marek Zak, Tomasz Sikorski, Szymon Krupnik, Magdalena Wasik, Katarzyna Grzanka, Daniel Courteix, Frederic Dutheil, Waldemar Brola

**Affiliations:** 1Jan Kochanowski University, Institute of Health Sciences, Collegium Medicum, Zeromskiego 5, 25-369 Kielce, Poland; wbrola@wp.pl; 2Jan Kochanowski University, Doctoral School, Zeromskiego 5, 25-369 Kielce, Poland; tomasz.sikorski@phd.ujk.edu.pl (T.S.); magdalena.wasik@phd.ujk.edu.pl (M.W.); katarzynagrzanka@o2.pl (K.G.); 3Symmetry, Medical Rehabilitation Centre, 41-208 Sosnowiec, Poland; krupny@gmail.com; 4Université Clermont Auvergne, Laboratory of the Metabolic Adaptations to Exercise under Physiological and Pathological Conditions (AME2P), 63000 Clermont-Ferrand, France; daniel.courteix@uca.fr; 5CHU Clermont-Ferrand, Occupational and Environmental Medicine, 63000 Clermont-Ferrand, France; fdutheil@chu-clermontferrand.fr; 6Université Clermont Auvergne, CNRS, LaPSCo, Physiological and Psychosocial Stress, 63000 Clermont-Ferrand, France

**Keywords:** virtual reality (VR) technology, telerehabilitation, exergaming, rehabilitation strategies, fall risk, elderly, seniors, OTAGO programme

## Abstract

Modern technologies are presently harnessed in response to a complex challenge of providing physiotherapeutic management in older adults. Fully immersive virtual reality (VR) solutions are acknowledged to viably enhance the overall effectiveness of traditional physiotherapeutic methods. A total of 60 community-dwelling older adults (over 75 years of age) were recruited for the study protocol. They were subsequently randomly split into four equally sized study groups (VR, CVR, OCULUS, and the classic programme group (OTAGO), and the physiotherapy sessions were pursued in the subjects’ homes for 3 weeks, 3 times a week, for 30 min in each group. At the first measurement point, respective study groups differed significantly in functional performance, as expressed in gait (POMA G) and individual static balance. The post hoc analysis indicated significantly higher scores in POMA G for the classic programme group vs. the results of the VR and CVR groups. On the other hand, the OCULUS group held significantly higher scores in individual balance and TUG, as compared to the other groups (*p* < 0.001). Making use of a virtual reality (VR) environment in the physiotherapeutic management of community-dwelling older adults appreciably enhanced individual functional performance, especially in terms of static balance. Physiotherapy management aided by VR technology solutions offers a viable alternative to traditional physiotherapeutic regimens (e.g., OTAGO programme) in enhancing individual functional performance. The innovatively self-designed VIRTUAL REALITY COMPREHENSIVE REHABILITATION ROOMS (VRCRR) solution may help out in pursuing a complex physiotherapy programme on an individual basis within one’s own home environment.

## 1. Introduction

As the world’s population keeps on aging at a dramatic pace, this poses a complex challenge that needs to be addressed in a no less complex, adequately structured manner [1,2]. This need holds especially true for the residents of various nursing facilities. Helping maintain individual wellbeing in an advanced age poses yet another hands-on challenge, both for the therapists and regular staff in such facilities [3,4,5].

Modern technology is well capable of offering a diversity of viable solutions, primarily in making innovative use of virtual reality (VR), which holds substantial, and yet unexplored potential in physiotherapeutic management [6]. The interactive, immersive VR systems may successfully be applied for post-stroke motor rehabilitation, post-cancer rehabilitation, cognitive assessment, and memory training [7,8,9,10]. Several brand-new, immersive VR devices (e.g., Oculus Rift and Carl Zeiss VR ONE plus) are already acknowledged to offer a real boost to the actual development of various VR applications [11,12,13,14].

It has already been established that nursing home residents, who are usually much disenchanted and frustrated with the conventional rehabilitation exercise regimens, quite enthusiastically welcome the introduction of various VR technology solutions [15]. Not only does this offer them a much welcome break from a tediously repetitive exercise routine, but also proves mentally engaging and extra stimulating. This is turn translates into much higher physical efficiency, combined with appreciable enhancement of individual cognitive capacity [16].

The present study has therefore set out to provide several key insights into the authors’ self-designed, innovative physiotherapy programme, aided by virtual reality (VR) technological solutions. It also compares it with several other rehabilitation options for the older adults affected by functional impairments. This innovative physiotherapy programme introduces a comprehensively structured procedure aimed at maintaining or enhancing individual balance and cognitive abilities within a fully immersive VR environment, specifically in the individuals over 75 years of age.

This is an innovative procedure, addressing four domains of individual functionality, i.e., static balance, cognitive abilities, dual-task functional activities, and aerobic training. These functionalities are already widely acknowledged to be absolutely essential for this group of older adults. The authors’ assertion is fully supported by numerous scientific societies and organisations dealing with overall prevention, as well as treatment management in seniors, e.g., BGS (British Geriatric Society), AGS (American Geriatric Society), ACSM (American College of Sports Medicine), and CDC (Centres for Disease Control and Prevention) [1,2].

Both the self-designed, comprehensive physiotherapy programme itself, as well as the actual platform for its hands-on implementation, offer a structured VR environment, comprised of several interconnected spaces (i.e., four function-driven rooms), specifically adapted for the pursuit of an innovatively self-designed, comprehensively structured physiotherapy programme.

The authors’ diligent review and detailed appraisal of current studies (PubMed—within the last 5 years—317 clinical studies) on physiotherapy and rehabilitation, whilst making use of various VR solutions, indicated them to be focused mostly on the stroke-induced upper limb dysfunction (174 clinical studies) [17], gait and balance disorders in Parkinson’s disease (39 clinical studies) [18], or on children affected by cerebral palsy (37 clinical studies) [19].

Given the above-referenced stratification of prevalent academic focus in the studies published to date, the present study offers a clearly pioneering approach to making comprehensive use of the VR technology. In fact, it is strongly focused on combining the advances of cutting-edge VR technology with their hands-on application, introducing an innovative approach to function-driven adaptation of the training VR environment.

It should also be highlighted at this juncture that a comprehensive physiotherapeutic programme, making use of several interconnected VR spaces (i.e., four function-driven rooms) clearly demonstrates that an appreciable enhancement of individual cognitive function, resulting from a radical modification of the rehabilitation environment itself, when combined with multi-component balance training and aerobic exercises regimen, may significantly improve individual functional performance.

The present study findings may well be regarded as a complete novelty in terms of the actual contribution of specific VR solutions to effectively enhance a comprehensively structured physiotherapy programme. It also has to be borne in mind at this point that a systematic literature review on VR applications in rehabilitation (dating back to 2020) still asserted that VR did not claim any superior efficacy in attenuating individual gait and balance deficits with regard to several medical conditions [20].

This particular assertion has now successfully been challenged by the authors, who set out to evaluate overall effectiveness of a self-designed physiotherapy programme, aided specifically by OCULUS Rift S and Carl Zeiss VR ONE plus devices, based on VR technology solutions, following their due adaptation to suit the specific requirements of the innovatively structured study protocol.

Not only did its application in the individuals over 75 years of age, affected by functional capacity impairment, prove successful in producing the most encouraging results in terms of appreciably enhanced individual functional performance, but also reinforced the authors’ belief that making use of such a complex, VR-aided protocol is the right way forward in making the rehabilitation of seniors a much more welcome therapeutic experience for all concerned.

## 2. Materials and Methods

### 2.1. Participants

A total of 95 community-dwelling older adults, residents of large cities, were recruited for the study; 60 individuals were pronounced fully eligible for attending the study protocol. Women made up 60% of the study group (36 subjects), and men 40% (24 subjects). Their mean age was 77.6 years.

The present study has been registered with the Australian New Zealand Clinical Trials Registry—Trial ID number: ACTRN12621000719831.

The procedure of having the study subjects allocated into the second stage was completed against the strict inclusion criteria, i.e., an informed written consent to attend the study protocol, age 75 years and over, doctor’s consent (a general practitioner or a geriatrician), MMSE ≥19, blood pressure <140–159 mmHg (systolic), and <90–99 mmHg (diastolic), gait speed >0.8 m/s, and BBS 42 and below.

Owing to an increased fall risk, the subjects with a BBS score of 42 and below this value were pronounced as affected by functional capacity impairment.

The exclusion criteria comprised visual impairment ≤2 dioptres, mean hearing loss ≤40–70 dB, functional limb shortening, Alzheimer’s disease, Parkinson’s disease, unstable cardiovascular disease, active cancer, no informed written consent to attend the study protocol, vertigo attacks (lasting more than 10 min per day—within the last 3 weeks prior to joining the study), GDS >8 points.

The subjects were subsequently split into 4 groups by computer-assisted randomization. A simple randomization was completed with the aid of MS EXCEL spread sheet. The calculation was carried out by way of using the (RAND) command for each subject, along with the allocation to respective study groups.

Randomization was completed by the principal investigator, whereas the double blinding was ensured through neither telling the subjects which specific physiotherapy programme they would follow, nor advising the attending physiotherapists which specific intervention would be implemented in each study group. The subjects in respective groups were characterised in terms of general health, functional ability, and sociodemographic data. Their characteristics are comprised in Table 1, whereas the recruitment procedure and the actual randomisation method are addressed in the flowchart (Figure 1).

### 2.2. Instruments

#### 2.2.1. The First Stage of Project Implementation—Preliminary Assessment

In the first stage, the subjects were assessed through making use of the following research tools:

Mini-mental state examination (MMSE)—developed to analyse the cognitive aspects of mental functioning [21].

Geriatric depression scale—15 items (GDS-15)—an abbreviated form of the GDS for screening, diagnosing, and assessing depression in older adults [22].

Instrumental activities of daily living (IADL)—applied to assess individual functional status by analysing the performance of activities of daily living embracing the selected 8 items [23].

Berg balance scale (BBS)—is a 14-item test applied to assess individual static balance and overall fall risk [24,25].

These tests were applied in the first stage of the study protocol in order to verify the subjects’ compliance with the adopted inclusion criteria, as well as establish their general health status.

#### 2.2.2. The Second Stage of Project Implementation—Evaluation of the Subjects Prior to the Commencement of Respective Physiotherapy Programmes

In the second stage, the subjects were assessed with the aid of the following research tools for a more complete assessment of individual balance, overall fall risk, gait, and postural functions (visuospatial functions):

Timed up and go test (TUG) [s]—designed to assess individual mobility, as well as exposure to overall fall risk, owing to certain stability disorders encountered when walking [26].

Timed up and go test cognitive (TUG_Cog_) [s]—based on carrying out the same scope of tasks as in the TUG test, but made more complex through having to count backwards from a number randomly selected (within the 20–100 range), when walking [27].

Timed up and go test manual (TUG_Man_) [s]—based on carrying out the same scope of tasks as in the TUG test, but made more complex through having to hold a cup of water, while walking [27].

A 10-m walk test (10 MW) [m/s]—a simple test, within the corridor tests series, which consists of walking the shortest distance, i.e., a distance of 10 m, during which functional changes can be noted. This test had been found to be positively correlated with an individual sense of balance [28].

Part one (POMA B)—consists of 9 tasks meant to assess individual balance while sitting, standing up, tripping with eyes open and closed, turning around, and sitting down. Part two (POMA—G) is used to assess individual gait against 7 factors, i.e., gait initiation, stride length and height, stride symmetry, gait continuity, gait path, and body posture [29].

Trail-making tests (TMT)—consists of two parts. In each part, 25 items had to be connected without lifting the pencil from the paper. Part A entailed connecting numbers (i.e., 1 to 2 to 3, etc.), while Part B entailed alternating the numbers with letters (i.e., 1 to A to 2 to B, etc.) [30].

Single-leg stance test (SLS)—designed to assess individual static balance. It is performed with eyes open (SLS OP) and eyes closed (SLS CL). The subject is supposed to stand on one leg without any third-party’s assistance, with the arms resting on the hips (aka arms akimbo) [31].

A 2 min step test (2MS)—designed to assess individual aerobic endurance. It was carried out in line with the methods proposed by Rikli and Jones [32].

#### 2.2.3. The Third Stage of Project Implementation—Post-Completion Assessment of the Physiotherapy Programmes

In the third stage, the post-completion assessment of the physiotherapy programmes was completed through making use of the research tools (tests and scales) referenced further above.

### 2.3. Study Design

The following test materials were used in our study:

Carl Zeiss VR ONE plus—The ZEISS^®^ VR ONE Plus contains two aspherical, biconvex lenses with +32.5 dioptres that facilitate fitting every subject within the display screen of a smartphone, which slides into the front tray of the head-mounted system at a distance of 44 mm, thus allowing a full 3D immersion.

Oculus Rift S—contains a number of sensors tracking the body movements. Thanks to the infrared LEDs and a camera placed on the goggles, it is possible to track the rotational movements of the head and pinpoint the position of the head in space. With the help of Touch VR controllers, carrying out various activities in virtual reality feels quite natural, just as in everyday life.

The physiotherapy management sessions were held for 3 weeks, 3 times a week, for 30 min, in each group, except for the first session, which lasted 60 min, so as to ensure that all participants were fully familiarised with how the programme actually worked. All sessions were carried out in the subject’s home environment (home, place of residence).

#### Physiotherapeutic Management

The physiotherapeutic treatment in each group was implemented by qualified physiotherapists boasting substantial (min. 2-year) hands-on clinical experience in working with older adults, inclusive of application of various VR devices, and training in a scope of dual-task activities.

**The Classic Programme group (OTAGO)**—a programme consisting of OTAGO exercise programme (OEP) exercises, embracing balance, flexibility, and resistance training. The balance exercises focused on standing, walking, walking up and down the stairs, and standing up from a chair. Rubber bands (e.g., Thera-Band^®^) were used for resistance exercises, whereas the stretching exercises complemented the regimen. The OTAGO exercise programme was implemented in line with the guidelines set down by its authors [33].

**The VR group**—(not) full immersion ensured by making use of Carl Zeiss VR One goggles, which “relocated” the subject into a virtual labyrinth in the “Maze Walk”, a publicly available application. This game was selected specifically with a view to creating a virtual reality environment within the study protocol [34]. Holding hands with the therapist throughout, the subject walked forward, backward, diagonally, and in a square. The subject also performed basic exercises in the form of a half-squat, by raising either one, or both upper limbs, making trunk twists, and bending sideways to both sides. Every 5 min, there was a 1 min break to avoid the risk of cybersickness (unwelcome dizzy spells) (Figure 2).

**CVR (DUAL-TASK + VR) group**—a programme comprised of dual-task exercises (15 min) and the application of Carl Zeiss VR One virtual reality (15 min). The dual-task exercises embraced cognitive tasks (1), repeating phrases while walking, (2) walking combined with adding by one, (3) walking combined with subtracting by one, and (4) walking while articulating a word chain (i.e., saying a word that begins with the last letter of the previous word). As training progressed, the subjects were challenged with longer phrases or adding/subtracting by more complex numbers (max. by 9), and structured motor tasks (1) walking with one or two balls (diameter = 20 cm) and (2) walking combined with kicking a basketball held in a net by the subject. The walking tasks covered walking forward, walking backward, and walking along the “S”-shaped route. The exercises were selected in line with the methods described in the study by Liu, 2017 [35]. The VR exercises covered the same scope of activities as in the VR group.

**The “OCULUS” group**—making use of the innovative VIRTUAL REALITY COMPREHENSIVE REHABILITATION ROOMS—(VRCRR), designed with the aid of Blender software (Blender software V. 2.93, Amsterdam, The Netherlands) as an additional component to be integrated with the physiotherapy programme. This pioneering and innovative package of four VRCCRs was self-designed and developed by the investigators, specifically with complex rehabilitation in mind.

Subsequently, the file had been transferred to UNREAL software package, and finally the image was converted to the Oculus glasses.

The VR environment—VIRTUAL REALITY COMPREHENSIVE REHABILITATION ROOMS—(VRCRR) in which the study subjects pursued their training procedure, assisted by the attending physiotherapist, is depicted in Figure 3.

Fifty individual models containing the four VRCRR were made (Figure 4). The cross-sectional photo below depicts the actual arrangement of the rooms on a map of virtual space.

The OculusRift is equipped with a number of controllers, which are used to guide the implementation of respective exercises with the special pointers. The VRCRR are functionally divided into the following zones: room 1—for cognitive exercises, room 2—for aerobic exercises (Figure 5), room 3—for static and dynamic balance exercises (Figure 6), room 4—for a scope of dual-task activities (Figure 7).

The first session lasted 60 min, including 30 min for getting familiarized with the test procedure and the Oculus Goggles, assisted by the physiotherapist (Figure 4). The following 30 min were dedicated to doing the actual exercises. The procedure called for visiting all four VRCRR in the pre-determined sequence. A 1.5 min break was scheduled between visiting respective rooms (Figure 8).

**Room no. 1** contained 12 white spherical shapes on which the letters of the alphabet (6) and numbers (6) were placed. The letters, each one the size of half the diameter of the spherical shape, were sequenced from A to F. The digits were sequenced from 1 to 6. They were randomly distributed in space, beyond the actual width of the visual field, so that the neck flexion and extension in the sagittal plane would not exceed 30 degrees. The study subject was supposed to carry out a sub-procedure divided into 3 stages, every 2 min, for 6 min.

Stage one comprised the recognition of 2 different combinations of spherical shapes (1 letter and 1 digit, e.g., A-1, B-2, C-3, D-4). The word chains (mixes) communicated to the subject had priorly been pre-recorded by the physiotherapist in a soundproofed room. There was a 3 s pause between them. Stage two comprised the recognition of 4 different combinations of spherical shapes (2 letters and 2 digits, e.g., A-1-B-2, C-2-D-3). Stage three comprised the recognition of 6 different combinations of spherical shapes (2 letters and 2 digits, e.g., A-1-B-2-C3, D-3-E-4-F-5). The study subject was supposed to indicate a specific mix, one by one, using the pointer located in his right hand.

**Room no. 2** had been designed with a scope of aerobic activities in mind. First, the subjects pursued some marching in situ, then pressed down a button on the controller which highlighted the arrow on the virtual reality map indicating a number of possible destinations of teleportation. The subjects had to operate this arrow in such a way as to fit within the demarcated corridor (marked in grey on the spatial map). The subjects performed 3 series of 1.5 min of marching in situ. The marching was carried out at the speed of 28 foot lifts. The subject’s feet were raised to the mid-shin height, so as to activate the quadriceps muscles.

**Room no. 3** contained 4 virtual obstacles in the form of horizontally positioned beams, each one at a different height. The first beam was positioned at the height of the ankle joint, the second one at the height of 1/3 of the distal part of the tibia. The fourth beam was positioned at the height of 1/3 of the proximal part of the tibia. The width of the beams in VR corresponded to 30 centimetres in reality. Each beam was to be crossed within a 1 min span. A 30 s resting time was allocated between respective beam crossings.

**Room no. 4** addressed a scope of dual-task training activities. The objective was to train the cognitive abilities. The scope of activities was supplemented with the Fukuda square. Eight black oval plates were created. Four letters and digits were randomly arranged on them. The subject, while listening to a mix consisting of 2 digits and 1 letter (e.g., 1A2), had to memorize them and then point to the specific digits and letters, one by one, while walking around the Fukuda square. Stepping accidentally on the line demarcating the square meant going back to the beginning of the exercise. The mixes consisted of 60 combinations, and the entire task was supposed to be completed within 5 min. One minute was allocated for rest.

### 2.4. Statistical Analysis

Statistical analysis was carried out with the aid of IBM SPSS v24 software. The sample size was calculated using a sample size calculator, whilst taking into account the age of the subjects, average number of free-living community dwellers who sustained at least one fall annually, and the likelihood of a static balance test result amounting to twice the standard deviation, assuming 80% test power. Overall consistency of the distribution was juxtaposed against the referential one with the aid of the Shapiro–Wilk test.

The measures of position (mean—$\bar{x}$) and the measures of dispersion (standard deviation—SD) were calculated. Differences between respective measurement points were calculated with the aid of the Wilcoxon test. The Kruskal–Wallis ANOVA test was used to calculate the differences between respective groups, depending on the actual measurement point. A significance level of α = 0.05 was assumed in the statistical analysis.

## 3. Results

At the first measurement point, the respective study groups differed significantly in functional performance, as expressed in gait (POMA G) and static balance, both with open and closed eyes. The post hoc analysis demonstrated significantly higher scores in POMA G for the classic group vs. the results of the VR and CVR groups. On the other hand, the OCULUS group had significantly higher scores in balance, both with the eyes closed and open, as compared to the other groups (*p* < 0.001). Functional performance in respective study groups, noted at the first measurement point, is addressed in Table 2.

Statistical analysis attested to the significant differences between respective study groups, as noted at the Second Measurement Point. The Oculus group had significantly higher scores in TUG, as compared to the VR and C groups (*p* < 0.001), as well as in static balance (with the eyes closed), as compared to the other groups. Group C showed significantly higher scores in POMA G, as compared to other groups. The CVR group had significantly higher scores in static balance (with the eyes open), as compared to other groups (*p* < 0.001). Functional performance at the Second Measurement Point is addressed in Table 3.

The C group demonstrated significantly lower differences, as noted between the two measurement points, in comparison with the other groups tested by POMA B, POMA total, and SLS OP. The OCULUS group attested to no differences between the VR group and the CVR group in the results of the tests specifically addressing balance. Table 4 highlights those differences in some detail.

## 4. Discussion

This study investigated the training effects of a specifically structured rehabilitation programme, aided by an innovative application of a self-designed and developed VR software package, dedicated for the VR OCULUS device. The programme aimed to aid physiotherapeutic intervention in older adults, based on the widely applied and acknowledged conventional methods of rehabilitation, made up of OTAGO exercises, in conjunction with a scope of dual-task activities, integrated with an innovative application of select VR devices.

The entire intervention scheme was therefore based on comprehensively addressing the four essential components that jointly stand for effective individual functionality, i.e., static balance, cognitive abilities, dual-task functional activities, and aerobic capacity. Considering that, jointly, they are widely acknowledged to directly correlate with functional performance in older adults, the authors principally focused their investigation on establishing to what extent a comprehensively structured, VR-aided study protocol might actually help out in turning a physiotherapy programme into an innovative, mentally far more engaging and stimulating experience for the seniors.

Our study demonstrates that a physiotherapy programme aided by VR technology solutions in the form of a Carl Zeiss VR One device, along with a more technologically advanced one like the OCULUS Rift S, may appreciably improve balance in older adults. Similar conclusions were reached by Kaminska et al. [36], who noted enhanced static and dynamic balance in older adults, as evidenced by the results of the dynamic gait index (DGI), tandem stance test (TST), and tandem walk test (TWT).

Likewise, Bainbridge et al. [37] achieved notable improvements in balance in older adults affected by balance disorders, as evidenced through the BBS scale, thus corroborating the overall effectiveness of such programmes as the extra aids to conventional rehabilitation methods. Even though the above-referenced studies made use of different devices than the present one (i.e., Kaminska used Xbox 360 Kinect, whereas Bainbridge—Nintendo Wii Fit), the general principle regarding the movements actually executed by the study subjects was the same as the one applied in the OCULUS goggles. In short, the environment within the virtual reality space generated specific stimuli which in turn forced specific movements upon the subjects, and their responsive interaction.

The present authors were inspired to include the OCULUS group in their own study by the one completed by Lubetzky et al. [38], who highlighted a certain potential for making efficient application of the OCULUS Rift device in the patients affected by vestibular deficits, exemplified on 25 subjects. Furthermore, Marchetto et al. [39] proposed that the kinematics of the head position, as measured by the OCULUS Rift, may offer a viable postural control tool, with no additional posturography equipment required.

In recent years, several studies making use of the VR goggles, e.g., Oculus Rift, have emerged. This is partly owed to the greater availability of those devices through more affordable prices, consequently translating into their wider application in medicine, as well as in rehabilitation, e.g., with stroke victims [39,40]. In their systematic review, Delgado et al. [41] drew on eight studies, four of which made use of VR head-mounted display (VR-HMD) technology as a tool to assess individual balance, and four that relied on it in their interventions. The authors concluded that it was not clear enough whether rehabilitation based exclusively on VR-HMD (i.e., not integrated with conventional therapeutic management) actually improved individual balance in the patients.

Our results imply that both the VR-HMD-based group (VR and OCULUS groups) as well as VR-HMD and dual-task activities (CVR) stand to improve balance in the patients affected by various balance disorders which directly translates into appreciably enhanced individual functional performance.

Although the most significant improvement in SLS OP scores was recorded by the CVR group making use of the VR environment combined with dual-task activities—by 23.6%, as compared to the first measurement point—the VR and OCULUS groups, with no benefit of a conventional physiotherapeutic regimen, also improved their balance scores on the SLS CL test, i.e., by 23.7% and 9.6%, respectively.

The application of the OCULUS device, as addressed in the published studies so far, was usually based on a single activity. Our own team of investigators developed and applied in the physiotherapy programme an innovatively structured solution, i.e., a set of four fully integrated, VIRTUAL REALITY COMPREHENSIVE REHABILITATION ROOMS (VRCRR), as graphically presented and described further above. In a sense, this is actually a pioneering effort as it offers an opportunity for the seniors to pursue their rehabilitation routine with the aid of a single physiotherapy programme, now, for the first time, also within a single, unbroken sequence with no intervening breaks, thus allowing them to focus entirely on completing the tasks at hand, merely by moving smoothly from one to the next.

This innovative solution facilitates working with a patient on four essential levels of functional performance within a single sequence, with no need to change the game or programme, effectively causing no disruption whatsoever in the ongoing flow of interactive tasks.

The patient, through moving from one room to another, whilst staying within the same virtual world generated by the programme, may, for example, freely switch between the exercises enhancing cognitive functions and the aerobic ones. In comparison with the games and programmes based on a single activity (i.e., concentrating on a single functionality), it offers a comprehensively structured therapy flow, with no distractions whatsoever, so the patient may remain perfectly focused at all times.

Effective application of the VRCRR package still requires further improvement by the authors, with a view to enabling higher performance scores in the patients. It seems only prudent to consider expanding the levels of difficulty, as this might reasonably translate into more demanding challenges for the patients in each one of the four VRCRR as their training continues. Substantial potential of the four VRCRR is clearly manifest with regard to appreciably enhancing the cognitive abilities in older adults.

The results of the Tinetti POMA test Total indicated improvement and no significant differences between the VR, CVR, and OCULUS groups, whereas the C group in which the intervention was based on OTAGO exercises evidenced the least improvement. The effect of the training regimen on information processing (as one of the key components of functional performance) was also assessed in this study. TMT A measures the subject’s psychomotor speed and visual scanning, whereas TMT B provides an insight into the working memory [42,43,44]. The best results in TMT B were achieved by the VR and OCULUS groups, with no statistically significant differences. For the TMT A test, the greatest improvement was observed in the VR group. Those outcomes may well translate into some potential for aiding individual cognitive abilities in more complex tasks making use of VR-HMD devices.

This offers the training environment particularly conducive to senso-motor, otolaryngological, and cognitive training. It is principally based on the fixation of stimuli, which the patient then converts from the virtual reality into the actual world environment, yet being denied a natural benefit of seeing his own body performing during the therapeutic intervention itself. The investigators also noted that it would make working with the patient much easier, if he could be hoisted above the floor by a purpose-designed harness.

Htut et al. [45] assert that the VR-based therapy offers a far more stimulating version of the rehabilitation exercises intended specifically for older adults, as opposed to the conventional exercise regimens. The authors’ own research also readily corroborates this assertion. All post-therapy individuals who had originally been allocated to the VR groups (VR, CVR, OCULUS) admitted that their “immersive” experience had been an absolute novelty to them, as well as a very stimulating and exiting one.

High-quality immersive experience offered by the VR devices, their cost-effectiveness, and overall accessibility are the main factors behind dynamically rising popularity of this cutting-edge technology. Even though it had originally been developed for pure entertainment purposes, it may now be harnessed, by no means less successfully, to aid rehabilitation of older adults, with a view to helping them recover their functional capacity, whilst enhancing their overall quality of life.

Appreciably encouraged by their promising experience with the application of VR immersive technology solutions in rehabilitating the seniors, the authors have sufficient grounds to believe this is indeed the right direction to be followed. This extremely versatile and adaptable technology is already acknowledged worldwide to be quite effective in aiding a diversity of physiotherapeutic strategies, especially those being developed with the institutionalised older adults in mind.

On the other hand, since there are no publications addressing such a widely structured application of VR solutions, as reflected in our own study protocol (i.e., four function-driven rooms and four different forms of activity), this renders any comparisons to be made between our own study outcomes and those reported by other investigators practically non-feasible. There is actually no common ground to have them juxtaposed. It is for this reason that we make references to those studies only that may offer, to some extent at least, a certain scope for comparing select aspects of respective research endeavours.

### 4.1. The Applied Character of the VR-Aided Physiotherapy Programme

By way of recapitulating, here are the key points highlighting the applied character of research into the VR-aided physiotherapy programme in older adults.

A structured physiotherapy programme, aided by purpose-adapted VR solutions dedicated specifically to the rehabilitation of older adults, offers a wide scope of mentally engaging, purpose-structured training activities, so that the specific skills acquired in the VR environment may be used throughout the activities of daily living and effectively remedy individual physical dysfunctions.Application of purpose-adapted VR solutions also proves quite beneficial in diminishing overall fall risk, especially in the therapeutic management of older adults. VR-aided physiotherapy programmes are widely regarded to be highly motivational for the participants, as well as well capable of restoring impaired motor abilities, balance, enhancing individual cognitive functions, and generally promoting overall self-reliance in an individual pursuit of basic activities of daily living.In view of a steadily growing share of advanced VR technologies in the medical services sector, overall cost-effectiveness of the VR-aided physiotherapy solutions is also considerable, as compared to the conventional physiotherapy methods requiring full-time staffing with highly trained, specialised personnel.The specifically structured VR solution (VRCRR), as presently proposed by the authors, also remains in full conformity with the European Commission’s long-term policy on healthy aging (www.age-platform.eu, accessed on 10 May 2022), and actually fleshes out the EU principle of lifelong learning in older adults (https://www.projects.aegee.org/educationunlimited/files/Lifelong_Learning_brief.pdf, accessed on 10 May 2022).In view of the widely acknowledged technological versatility of VR technology, there is also a substantial potential for the application of physiotherapy programmes aided by purpose-adapted VR solutions on an individual basis, in the patients’ own home environment, even without the attending physiotherapist on hand. His supervision may well be implemented online, by way of making use of the real-time data collection offered by the training outcome tracking software, i.e., one of the standard VR options.

### 4.2. Study Limitations

This study is not free from its limitations, however. The first one consists of the fact that even though the study protocol envisaged four groups of subjects, they were far from numerous. As far as the future studies are concerned, however, the authors intend to focus on boosting the size of their study groups by at least two-fold (up to 30 individuals each), or alternatively have them reduced to just two. The first study group would be assigned a combination of conventional rehabilitation exercises, e.g., the OTAGO type, aided by the select VR devices, whereas the other one would have its exercises aided by OCULUS Rift, with no conventional exercises included in the training regimen.

Furthermore, the authors acknowledge that non-performance of the tests at the second and third measurement points, in random order, may have affected some of the test results of some, as, for instance, carrying out the TMT B immediately after TMT A may affect the actual duration of their respective execution, much owing to the similarity of their structural components.

The other limitation was that during the VR-HMD therapy, some of the subjects occasionally experienced bouts of cybersickness (unwelcome dizzy spells).

Even though the authors had established dizziness of neurological origin as a rigid exclusion criterion, and had pre-emptively asked their subjects to have their vagus response assessed following their transfer to a virtual environment, i.e., prior to the actual commencement of the therapy, the actual study outcomes indicated that a more precise structuring of the exclusion criteria would be required to minimise an occasional incidence of dizzy spells among the subjects, when the study protocol was already in progress.

Approx. 1/3 of the study subjects reported dizziness during 1–5 training sessions aided by VR-HMD. Despite having been advised they could opt out of the study protocol altogether, none of the subjects did so, and all keenly continued exercising after a few minutes’ rest. The authors did make a working assumption this may well have influenced some of the study outcomes, however.

## 5. Conclusions

Making use of Virtual Reality (VR) environments in the physiotherapeutic management of community-dwelling older adults appreciably enhanced individual functional performance, especially in terms of static balance.Physiotherapy management aided by VR technology solutions offers a viable alternative to conventional physiotherapeutic regimens (e.g., OTAGO programme) in enhancing individual functional performance.The innovatively self-designed VIRTUAL REALITY COMPREHENSIVE REHABILITATION ROOMS (VRCRR) solution may help out in pursuing a complex physiotherapy programme on an individual basis within one’s own home environment.

## Figures and Tables

**Figure 1 ijerph-19-06018-f001:**
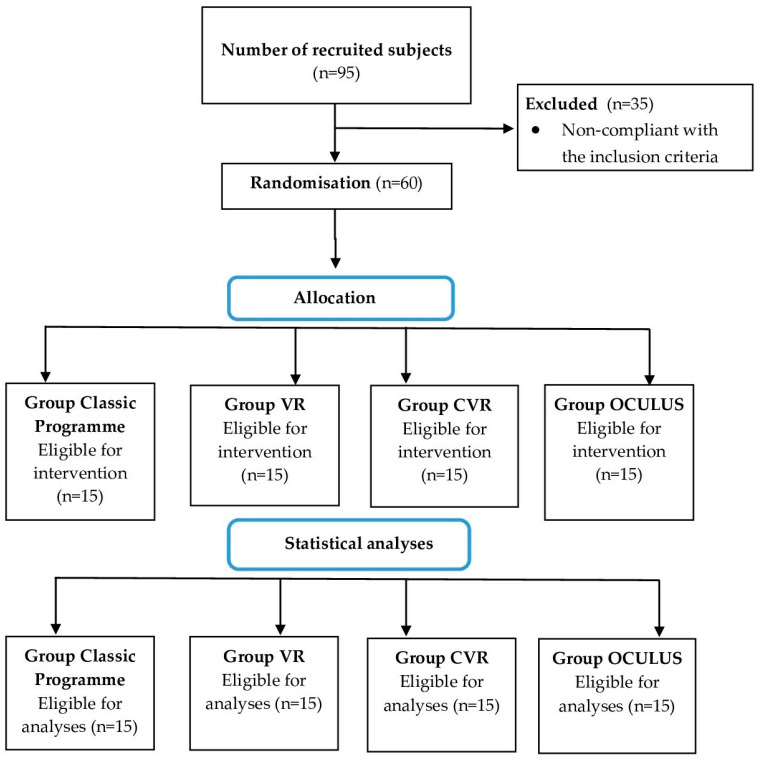
Recruitment process and the sizes of respective study groups.

**Figure 2 ijerph-19-06018-f002:**
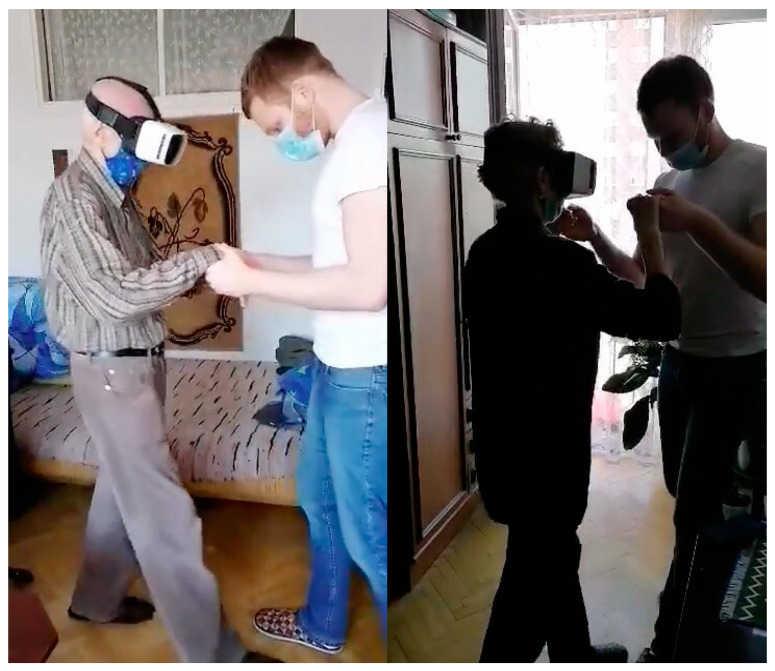
Therapy session aided by Carl Zeiss VR ONE plus device.

**Figure 3 ijerph-19-06018-f003:**
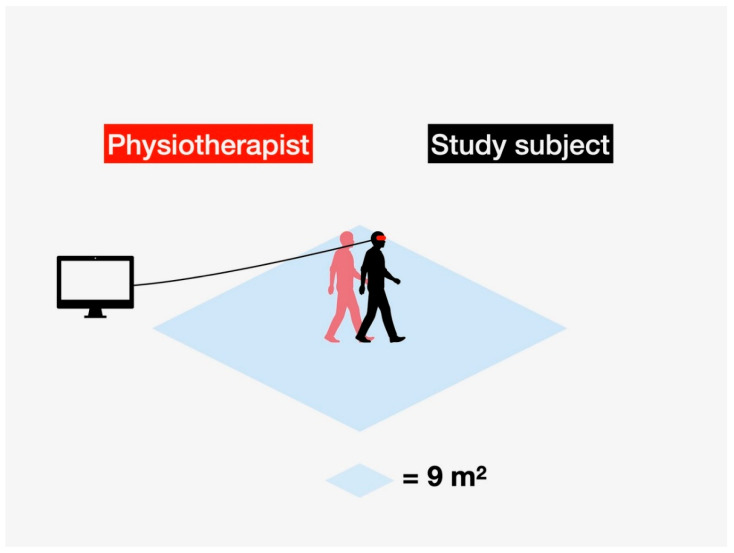
VIRTUAL REALITY COMPREHENSIVE REHABILITATION ROOMS—(VRCRR)—an area where the physiotherapy intervention was carried out in the study subjects, assisted throughout by the attending physiotherapist.

**Figure 4 ijerph-19-06018-f004:**
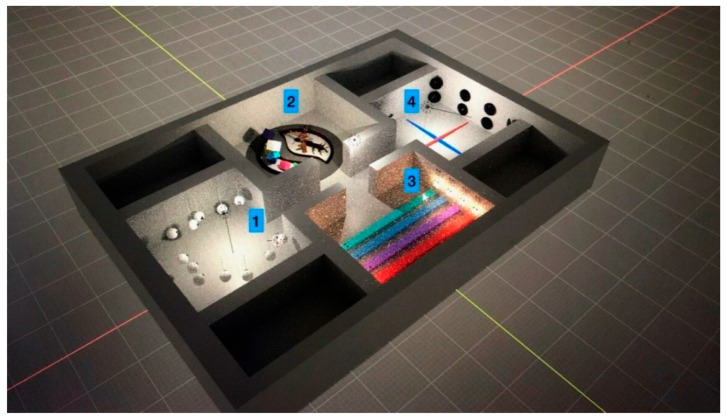
Spatial view of the VIRTUAL REALITY COMPREHENSIVE REHABILITATION ROOMS (VRCRR).

**Figure 5 ijerph-19-06018-f005:**
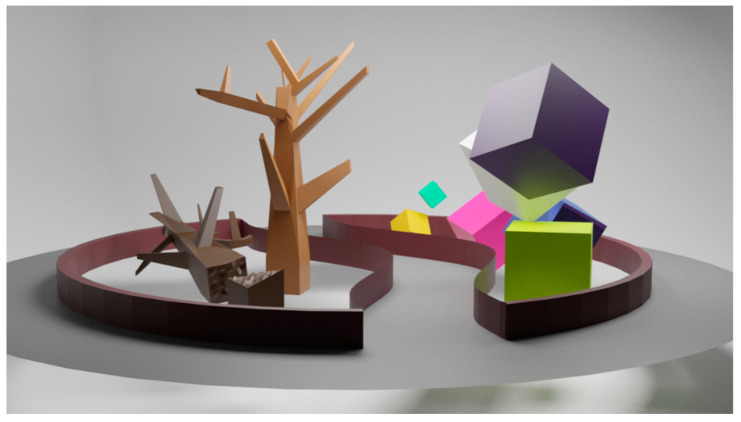
Room no. 2 designed for a scope of aerobic activities.

**Figure 6 ijerph-19-06018-f006:**
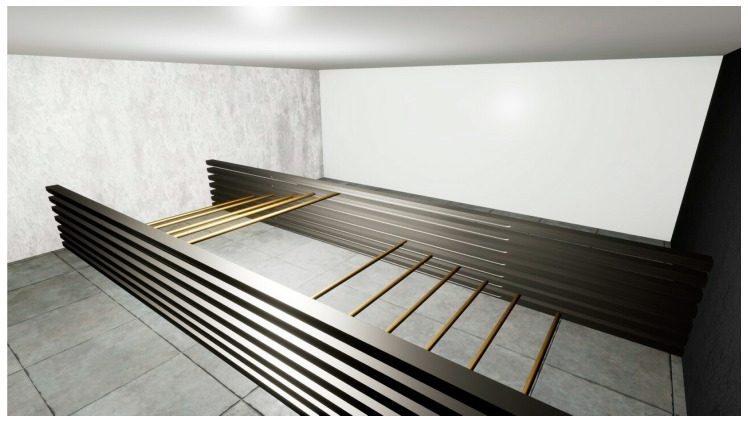
Room no. 3 designed for static and dynamic balance exercises.

**Figure 7 ijerph-19-06018-f007:**
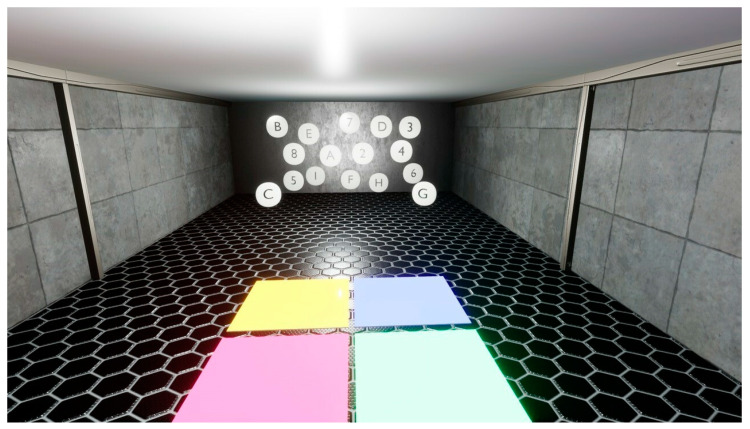
Room no. 4 designed for a scope of dual-task activities.

**Figure 8 ijerph-19-06018-f008:**
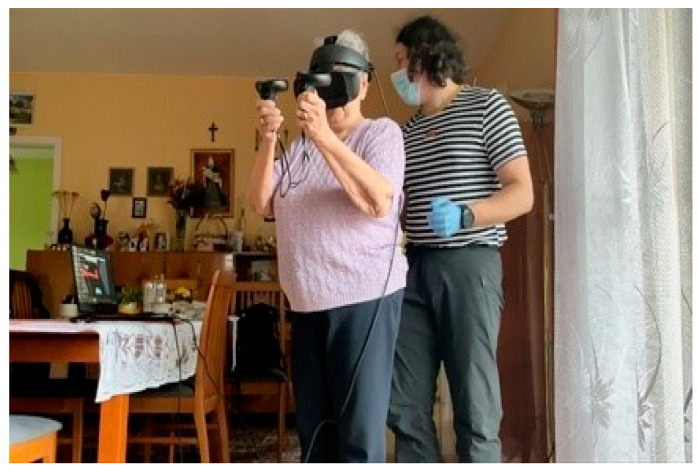
The VR-based intervention making use of the Oculus goggles, assisted by a physiotherapist.

**Table 1 ijerph-19-06018-t001:** Characteristics of respective study groups.

Group	Group Classic Programme *N* = 15		Group VR *N* = 15		Group CVR *N* = 15		Group OCULUS *N* = 15		ANOVA KW	
Measurement	$\bar{x}$	SD	$\bar{x}$	SD	$\bar{x}$	SD	$\bar{x}$	SD	F(3)	*p*
Age [years]	76.66	1.63	76.66	1.5	78.06	3.73	79.13	3.56	9.6	0.08
Body weight [kg]	72.86	13.7	71.2	12.75	70.2	10.3	71	10.74	0.24	0.99
Height [cm]	1.66	1.82	1.67	1.85	1.63	1.74	1.65	1.8	1.35	0.90
BMI [kg/m^2^]	26.25	3.47	25.32	2.22	26.33	4.04	26.03	4.07	1.93	0.97
Education (%)									χ^2^	*p*
Primary	46.7		66.7		33.3		40		3.75	0.74
Vocational	13.3		20		40		26.7		3.11	0.84
Secondary	26.7		26.7		13.3		33.3		1.68	0.98
Higher	6.7		0		0		6.7		2.06	0.96
Falls sustained within the last year (YES, %)	20		6.7		46.7		46.7		8.57	0.03
Number of medications taken regularly	5.13 (2.06)		4.46 (2.5)		3.33 (1.71)		3.53 (1.50)		7.19	0.28
MMSE (pts)	26.33 (2.09)		27.53 (1.45)		27.8 (1.52)		28.26 (1.1)		8.59	0.16
GDS-15 (pts)	4.42 (1.6)		4.67 (1.54)		4.73 (1.71)		5.33 (1.95)		1.39	0.99
IADL (pts)	18.33 (1.95)		19.73 (1.87)		19.26 (1.38)		20.26 (2.25)		6.76	0.08
BBS (pts)	39.33 (1.34)		40.33 (1.67)		40 (4.05)		40.53 (4.48)		2.42	0.96

Abbreviations: $\bar{x}$—mean, SD—standard deviation, KW—Kruskal–Wallis test, χ^2^—Pearson chi-square test, BMI—body mass index, MMSE—mini-mental state examination, GDS-15—geriatric depression scale—15 items, IADL—instrumental activities of daily living, BBS—Berg balance scale.

**Table 2 ijerph-19-06018-t002:** Functional performance per respective study groups, as noted at the first measurement point.

Group	Group Classic Programme *N* = 15		Group VR *N* = 15		Group CVR *N* = 15		Group OCULUS *N* = 15		ANOVA KW	
Measurement	$\bar{x}$	SD	$\bar{x}$	SD	$\bar{x}$	SD	$\bar{x}$	SD	F(3)	*p*
TUG [s]	13.29	0.97	13.51	0.99	14.75	2.49	15.61	1.72	14.29	0.01
TUG_Cog_ [s]	15.85	1.2	16.45	1.61	20.59	8.38	19.04	6.94	3.77	0.81
TUG_Man_ [s]	13.62	1.18	13.36	1.14	13.48	1.61	13.98	1.95	0.42	0.93
10MW [m/s]	1.06	0.04	1.07	0.06	1.06	0.11	1.07	0.08	0.23	0.97
POMA B [pts]	10.66	0.9	10.6	1.12	10.53	1.06	10.4	0.73	0.61	0.89
POMA G [pts]	9.6	0.63	7.73	1.33	7.8	1.20	8.26	1.1	21.47	<0.001
POMA Total [pts]	20.26	1.03	18.33	1.63	18.33	1.49	18.66	1.29	17.75	0.002
TMTA [s]	66.7	10.36	66.32	12.67	71.81	13.67	69.23	12.58	2.19	0.97
TMTB [s]	148.83	25.27	141.62	20.7	134.4	25.79	146.7	21.56	3.45	0.86
SLS OP [s]	5.35	1.37	5.61	2.45	20.2	17.27	14.39	4.15	33.96	<0.001
SLS CL [s]	1.451	0.74	1.18	1.36	4.05	5.31	8.45	1.79	30.99	<0.001
2MS [number of lifts]	55.2	16.69	60.4	13.56	53.13	11.37	60.86	11.3	3.32	0.87

Abbreviations: $\bar{x}$—mean, SD—standard deviation, KW—Kruskal–Wallis test, TUG—timed up and go, TUG_Cog_—timed up and go cognitive, TUG_Man_—timed up and go manual, 10 MW—10-m walk test, Poma B—balance, Poma G—gait, Poma Total—Tinetti performance-oriented mobility assessment, TMT A—trail-making test A, TMT B—trail-making test B, SLS OP—single-leg stance open eyes, SLS CL—single-leg stance closed eyes, 2MS—two-minute step test.

**Table 3 ijerph-19-06018-t003:** Functional performance per respective study groups, as noted at the Second Measurement Point.

Group	Group Classic Programme *N* = 15		Group VR *N* = 15		Group CVR *N* = 15		Group OCULUS *N* = 15		ANOVA KW	
Measurement	$\bar{x}$	SD	$\bar{x}$	SD	$\bar{x}$	SD	$\bar{x}$	SD	F(3)	*p*
TUG [s]	11.89	0.8	12.65	1.01	14.19	2.52	14.42	1.81	19.3	0.001
TUG_Cog_ [s]	15.07	1.03	15.53	1.5	19.66	7.94	17.55	6.32	1.71	0.63
TUG_Man_ [s]	12.41	0.92	13.13	2.03	12.87	1.43	12.4	1.46	2.24	0.97
10MW [m/s]	1.13	0.05	1.08	0.07	1.1	0.09	1.10	0.08	3.87	0.8
BBS [pts]	41	1.13	41.33	1.44	40.6	4.13	41.26	10.73	1.91	0.98
POMA B [pts]	11.06	0.79	11.06	0.88	10.86	0.99	10.73	0.59	1.73	0.99
POMA G [pts]	9.73	0.45	8.26	1.43	8.93	1.28	9.06	0.79	11.42	0.04
POMA Total [pts]	20.86	0.83	19.46	1.30	19.8	1.56	19.8	1.01	10.34	0.07
TMTA [s]	62.94	8.98	58.72	9.4	69.90	11.65	65.40	10.11	8.33	0.18
TMTB [s]	140.43	23.54	122	20.60	127.62	21.94	135.23	20.15	5.10	0.59
SLS OP [s]	6.70	1.89	5.75	2.68	24.98	18.49	16.44	2.69	41.77	<0.001
SLS CL [s]	1.55	0.75	1.46	1.51	5.06	5.21	9.26	1.94	33.04	<0.001
2MS [number of lifts]	58.6	16.25	64.6	12.84	56.6	11.77	64.06	11.24	4.02	0.25

Abbreviations: $\bar{x}$—mean, SD—standard deviation, KW—Kruskal–Wallis test, TUG—timed up and go, TUG_Cog_—timed up and go cognitive, TUG_Man_—timed up and go manual, 10 MW—10-m walk test, BBS—Berg balance scale, Poma B—balance, Poma G—gait, Poma Total—Tinetti performance-oriented mobility assessment, TMT A—trail-making test A, TMT B—trail-making test B, SLS OP—single-leg stance open eyes, SLS CL—single-leg stance closed eyes, 2MS—two minute step test.

**Table 4 ijerph-19-06018-t004:** The differences in the tests specifically addressing balance, as noted between respective measurement points.

Group	Group Classic Programme *N* = 15		Group VR *N* = 15		Group CVR *N* = 15		Group OCULUS *N* = 15	
Measurement	$\bar{x}$	SD	$\bar{x}$	SD	$\bar{x}$	SD	$\bar{x}$	SD
BBS [pts]	−1.67	−0.21	−1	−0.22	−0.6	0.08	−0.73	6.24
POMA B [pts]	−0.11	−0.1	−1.07 *	−0.23	0.3 *	−0.07	0.31 *	−0.14
POMA G [pts]	−1.7	−0.17	−0.5	0.1	−1.1	0.07	−0.31	−0.30
POMA Total [pts]	−0.6	−0.19	−1.1 **	−0.33	−1.5 **	0.07	−1.1 **	−0.27
SLS OP [s]	−0.51	0.55	−0.1	0.22	−2.1 ***	1.21	2.1 ***	−1.46

Abbreviations: $\bar{x}$—mean, SD—standard deviation, BBS—Berg balance scale, Poma B—balance, Poma G—gait, Poma Total—Tinetti performance-oriented mobility assessment, SLS OP—single-leg stance open eyes, *t*—Student *t*-test. POMA G * *t* = 14.43, * *p* < 0.0001 C vs. VR, *t* = 5.98, *p* < 0.0001 C vs. CVR, * *t* = 4.42, *p* = 0.0001 C vs. OCULUS. POMA total ** *t* = 5.01, *p* < 0.0001 C vs. VR, *t* = 16.47, *p* < 0.0001 C vs. CVR, *t* = 5.67, *p* < 0.0001 C vs. OCULUS. SLS OP *** *t* = 4.59, *p* = 0.0001 C vs. CVR, *t* = 3.94, *p* = 0.0005 C vs. OCULUS.

## Data Availability

The datasets generated and/or analysed during the current study are available from the corresponding author upon reasonable request.

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
