# Peer review of "Physiotherapy Programmes Aided by VR Solutions Applied to the Seniors Affected by Functional Capacity Impairment: Randomised Controlled Trial"

_ijerph, 2022, doi:10.3390/ijerph19106018_

Round 1

Reviewer 1 Report

With this study, the authors aimed to assess the effectiveness of physiotherapeutic management based on VR technology solutions for seniors over 75 years. It is an interesting and worth to be studied issue.

Introduction

The introduction is relatively well written. However, it could report better previous results.

Lines 71-73 Please add the reference.

Methods

Please refer to the subjects' eligibility in the participants' section.

All tables must be formatted to increase legibility.

It is not clear to me why the results of the berg balance scale were not included in table 1.

When describing the instruments, the rationale for the tests used should be explained.

In the second and third stages of Project evaluation please refer to the order of tests application. In the case no random order was used, consider the possibility of being a study limitation.

Line 212 correct a virtual

Results

Are presented satisfactorily.

Discussion

Is presented satisfactorily.

Conclusion

Is presented satisfactorily.

References

Are presented satisfactorily.

Author Response

Authors' responses to the Reviewers

The original title of the study:

Title:Physiotherapeutic management aided by VR technology vs. conventional methods, in seniors over 75 years of age affected by impaired functional capacity: a Randomised Controlled Trial

has been amended with the following wording:

Physiotherapy programmes, aided by VR solutions, applied to the seniors affected by functional capacity impairment: Randomised Controlled Trial 

Journal: International Journal of Environmental Research and Public Health 

Special Issue“Physiotherapy and Rehabilitation as Modern-Day Medical Challenge in Older Adults

             Submission ID: ijerph-1668850

The Authors have diligently addressed all the concerns raised by the Reviewers. Hopefully, the revised version of the manuscript will merit the Reviewers' satisfaction.

AUTHORS' RESPONSES TO THE REVIEWER'S RECOMMENDATIONS

REVIEWER # 1

Comments and Suggestions for Authors

With this study, the authors aimed to assess the effectiveness of physiotherapeutic management based on VR technology solutions for seniors over 75 years.

It is an interesting and worth to be studied issue.

Introduction

The introduction is relatively well written. However, it could report better previous results.

AUTHORS' RESPONSE:

In line with the Reviewer's recommendations, the INTRODUCTION section has been radically re-edited and supplemented accordingly.

Lines 71-73 Please add the reference.

AUTHORS' RESPONSE:

Pertinent reference has been supplemented accordingly.

Methods

Please refer to the subjects' eligibility in the participants' section.

AUTHORS' RESPONSE:

The inclusion/exclusion criteria had originally been in the 2.3. Study Design sub-section. They have now been moved to the Participants sub-section to enhance overall clarity of the manuscript.

All tables must be formatted to increase legibility.

AUTHORS' RESPONSE:

In line with the Reviewer's recommendation, all tables have now been reformatted to increase their overall clarity; pertinent numerical data have been unified to 2 decimal points throughout.

It is not clear to me why the results of the berg balance scale were not included in table 1.

AUTHORS' RESPONSE:

In compliance with the adopted inclusion criteria, the subjects were supposed to have scored below 42 points on the Berg scale.

The Authors also concede the Reviewer's point and have therefore moved the pertinent Berg test scores to Table 1, to supplement general characteristics of respective study groups.

When describing the instruments, the rationale for the tests used should be explained.

AUTHORS' RESPONSE:

In line with the Reviewer's recommendation, this has now been clarified accordingly.

In the second and third stages of Project evaluation please refer to the order of tests application. In the case no random order was used, consider the possibility of being a study limitation.

AUTHORS' RESPONSE:

In line with the Reviewer's recommendation, the Authors have supplemented the Study Limitations sub-section accordingly.

Line 212 correct a virtual

AUTHORS' RESPONSE:

Regretfully, the Reviewer's concern has not been identified in the manuscript.

No Authors' response, then.

Results

Are presented satisfactorily.

Discussion

Is presented satisfactorily.

Conclusion

Is presented satisfactorily.

References

Are presented satisfactorily.

Submission Date

21 March 2022

Date of this review

31 Mar 2022 01:08:55

______________________________________________________________

Reviewer 2 Report

Overall this research is very interesting and was conducted in a well-organized manner. The paper is well-written; a good read. However, there are some weak spots that either need a more elaborate clarification or a better justification of choices.

It seems that title too long, therefore I suggest authors to change the title to e.g.,  "The application of VR in physiotherapeutic management for seniors with impaired functional capacity: a Randomized Controlled Trial" or something like that. 

Introduction

  • The introduction seems too short and includes unnecessary statements which are not required. It is better if point to point statement is made related to the objective of the study.
  • Line 59, 65: Authors should be cautious when use the term interactive and immersive VR systems because because many systems are interactive but not immersive.
  • The introduction missing motivation to use VR in rehabilitation, so I felt that first two paragraph of discussion better to be in the introduction section instead. 
  • Moreover, introduction seems to be too concise with the inability to summarize the authors’ contributions, it would be also better to have hypothesis or research questions in which authors could improve introduction section. 
  • I would suggest authors to change condition names into something else instead of using brand names like OCULUS and Carl Zeiss.  

Materials and methods

  • The session were 3 weeks, 3 times a week for 30 minutes, I wonder when the authors did the statistical analysis why did they polled the data without having a look at the differences over weeks to see the effects developed.
  • How many trials were for each condition?
  • The walking task is not obvious and the tracking area is not specified as well, what was the safety aspect (hitting the wall or an object in the room) for participant here? It would be nice if this will be great if illustrated with figure. 
  • Dual task accompanied with walking task was not clarified and authors could elaborate more to make it clear for the readers. E.g., what was the time between each stimuli when presented to participants? The best suggestions for authors could I make to improve explanation of the dual task with visualized figures. 
  • What was the dimensions of the virtual environment? 
  • What was the latency of the VR system that could have an effect on results of study? 
  • It would be nice if authors include visual representation of their results along with tables.  

Discussion 

  • The discussion seems to be too long with the inability to summarize the authors’ point of view. The authors could not provide interpretation or justification for the results, therefore it discussion should provide explanation why VR training is better than conventional therapy for instance. 
  • What are implications could this study hold for the future of physiotherapy, or insights could this research make for the community-dwelling elderly people. 
  • In the study limitations section I see some intention to explore cybersickness in which authors report some participants have symptoms. 
  • Why authors did not consider involving standard questionnaires to measure presence and simulator sickness. See the SUS PQ
    [1] and the SSQ [2], for instance.
    [1] M. Usoh, E. Catena, S. Arman, and M. Slater. Using Presence
    Questionaires in Reality. Presence: Teleoperators & Virtual Environments, 9(5):497-503, 1999.
    [2] R. Kennedy, N. Lane, K. Berbaum, and M. Lilienthal. Simulator
    Sickness Questionnaire: An Enhanced Method for Quantifying Simulator Sickness. The International Journal of Aviation Psychology, 3(3):203-220,
    1993.

Conclusion

  • The conclusion is too arbitrary. Suggest rewriting the conclusion.
  • Especially, Nr. 1 for instance, reported VR findings in physiotherapy management of older adults, which have been published already many times long time ago. 

Author Response

Authors' responses to the Reviewers

The original title of the study:

Title:Physiotherapeutic management aided by VR technology vs. conventional methods, in seniors over 75 years of age affected by impaired functional capacity: a Randomised Controlled Trial

has been amended with the following wording:

Physiotherapy programmes, aided by VR solutions, applied to the seniors affected by functional capacity impairment: Randomised Controlled Trial 

Journal: International Journal of Environmental Research and Public Health 

Special Issue“Physiotherapy and Rehabilitation as Modern-Day Medical Challenge in Older Adults

             Submission ID: ijerph-1668850

The Authors have diligently addressed all the concerns raised by the Reviewers. Hopefully, the revised version of the manuscript will merit the Reviewers' satisfaction.

AUTHORS' RESPONSES TO THE REVIEWER'S RECOMMENDATIONS

REVIEWER # 2

Comments and Suggestions for Authors

Overall, this research is very interesting and was conducted in a well-organized manner. The paper is well-written; a good read. However, there are some weak spots that either need a more elaborate clarification or a better justification of choices.

It seems that title too long, therefore I suggest authors to change the title to e.g., 

The application of VR in physiotherapeutic management for seniors with impaired functional capacity: a Randomized Controlled Trial" or something like that. 

AUTHORS' RESPONSE:

In line with the Reviewer's recommendation, the manuscript's title has been altered accordingly.

Introduction

  • The introduction seems too short and includes unnecessary statements which are not required. It is better if point to point statement is made related to the objective of the study.

AUTHORS' RESPONSE:

In line with the Reviewer's recommendation, the Introduction section has been radically edited and supplemented accordingly, with a view to enhancing its overall clarity.

  • Line 59, 65: Authors should be cautious when use the term interactive and immersive VR systems because many systems are interactive but not immersive.

AUTHORS' RESPONSE:

The Authors should like to point out that the terms indicated by the Reviewer had not been used interchangeably throughout the manuscript, so no amendments are required in this regard.

The introduction missing motivation to use VR in rehabilitation, so I felt that first two paragraph of discussion better to be in the introduction section instead. 

AUTHORS' RESPONSE:

The Introduction section has been supplemented accordingly.

  • Moreover, introduction seems to be too concise with the inability to summarize the authors’ contributions, it would be also better to have hypothesis or research questions in which authors could improve introduction section. 

AUTHORS' RESPONSE:

In line with the Reviewer's recommendation, the Introduction section has been supplemented and expanded accordingly.

  • I would suggest authors to change condition names into something else instead of using brand names like OCULUS and Carl Zeiss.  

AUTHORS' RESPONSE:

The Authors opted for the brand names of the VR devices applied in their study, specifically with aiding the search in the databases in mind. Besides, the Authors also wanted to preclude any potential confusion between them.

Materials and methods

  • The session were 3 weeks, 3 times a week for 30 minutes, I wonder when the authors did the statistical analysis why did they polled the data without having a look at the differences over weeks to see the effects developed.

AUTHORS' RESPONSE:

This is going to be the subject matter of the Authors' subsequent studies. The variability of results between respective weeks would thus indicate the learning effect. It would be beneficial to establish how fast the study subjects would get used to so many stimuli. This speed is directly correlated with the learning effect.

Already back in 2011, the team headed by Takiyama (Nature Proceedings), demonstrated that neuronal redundancy in the motor cortex was related to learning speed. The more neurons working simultaneously, the faster the learning speed gains. Thus, in the case of VR solutions, learning to directionally amplify the neuronal signals is going to be the main subject in focus in the Authors' further studies.

Takiyama, K., Okada, M. Maximization of learning speed in motor cortex due to neuron redundancy. Nat Prec(2011).

  • How many trials were for each condition?

AUTHORS' RESPONSE:

Provided the Authors understand the Reviewer correctly, the functional tests are being meant.

The Authors would also like to confirm that prior to completing each specific test, a dry run was made to ensure full compliance.

  • The walking task is not obvious and the tracking area is not specified as well, what was the safety aspect (hitting the wall or an object in the room) for participant here? It would be nice if this will be great if illustrated with figure. 

AUTHORS' RESPONSE:

The test was carried out in full compliance with applicable safety rules for Zaiss One and Oculus Rift S devices, respectively. The figure further below depicts how those safety constraints were applied in practice. The safe space in virtual reality environment is approx. 4 m2, whereas the Authors carried out their testing procedure within the space of 9 m2.

  • Dual task accompanied with walking task was not clarified and authors could elaborate more to make it clear for the readers. E.g., what was the time between each stimuli when presented to participants? The best suggestions for authors could I make to improve explanation of the dual task with visualized figures.

AUTHORS' RESPONSE:

Figure 3 actually addresses the manner of carrying out this scope of activity.

  • What was the dimensions of the virtual environment? 

AUTHORS' RESPONSE:

The safe space in virtual reality environment is approx. 4 m2 The safe space (i.e. the outline inside the VR environment was the outline recommended by the manufacturer of the VR device (this system is called Guardian), defined by six steps of unrestricted movement (i.e. forward, backward, up, down, left and right, along with tracking the rotational head movements. Each one of the training VR rooms had the actual floor space of approx. 16 m2).

  • What was the latency of the VR system that could have an effect on results of study? 

AUTHORS' RESPONSE:

The Authors are happy to confirm that whatever latency characterised the VR system, it had no direct bearing on the actual execution of the training procedure itself.

The Authors further assume that its value must have been negligible, as it did not interfere in any way with overall smoothness of completing the successive training tasks by the study subjects.

The Authors did not get any negative feedback on this variable from the attending physiotherapists, either.

This piece of system performance data may most likely be extracted from the manufacturer's technical specification, though.

  • It would be nice if authors include visual representation of their results along with tables.

AUTHORS' RESPONSE:

Owing to the publisher's layout restrictions in place, the Authors are unable to have the Reviewer accommodated on this particular score. The Authors also believe that given the choice, the tabularised format of vital data presentation renders itself much better to unequivocal interpretation, effectively leaving no margin for errors.

Discussion 

  • The discussion seems to be too long with the inability to summarize the authors’ point of view. The authors could not provide interpretation or justification for the results, therefore it discussion should provide explanation why VR training is better than conventional therapy for instance. 

AUTHORS' RESPONSE:

The Discussion section has diligently been re-edited, with a view to enhancing, as well as boosting the Authors' key message.

  • What are implications could this study hold for the future of physiotherapy, or insights could this research make for the community-dwelling elderly people. 

AUTHORS' RESPONSE:

The Authors addressed this shortcoming in the Study Limitations section accordingly.

  • In the study limitations section I see some intention to explore cybersickness in which authors report some participants have symptoms. 

AUTHORS' RESPONSE:

The Authors concede the Reviewer's point that in the present study this should have been ensured, so it was rightly branded a legitimate shortcoming that the Authors failed to have done so.

  • Why authors did not consider involving standard questionnaires to measure presence and simulator sickness.

AUTHORS' RESPONSE:

The Authors also plan to make use of the suggested questionnaires in their future projects on the application of VR solutions, so as to take due note of the issue.

See the SUS PQ
[1] and the SSQ [2], for instance.
[1] M. Usoh, E. Catena, S. Arman, and M. Slater. Using Presence
Questionaires in Reality. Presence: Teleoperators & Virtual Environments, 9(5):497-503, 1999.
[2] R. Kennedy, N. Lane, K. Berbaum, and M. Lilienthal. Simulator
Sickness Questionnaire: An Enhanced Method for Quantifying Simulator Sickness. The International Journal of Aviation Psychology, 3(3):203-220,
1993.

Conclusion

  • The conclusion is too arbitrary. Suggest rewriting the conclusion.
  • Especially, Nr. 1 for instance, reported VR findings in physiotherapy management of older adults, which have been published already many times long time ago. 

AUTHORS' RESPONSE:

Even though the Reviewer was quick to point out that numerous academic reports on the application of VR technology in the physiotherapeutic management of older adults had indeed been published to date (i.e. the fact never disputed by the Authors), it would appear that the key merits of the present study, e.g. an innovatively self-designed and structured VR application, i.e. four function-driven, comprehensive rehabilitation rooms (VRCRR), has somehow eluded him altogether.

Whilst revising the manuscript, the Authors ensured that the key merits of the present study protocol have now been suitably highlighted, so that all the major differences with the findings yielded by the studies published to date in this domain are now far easier to appreciate.

Hopefully, the Reviewer might also be ready to have them eventually acknowledged.

Reviewer 3 Report

REVIEW

Manuscript number ijerph- 1668850

Title: Physiotherapeutic management aided by VR technology vs. conventional methods, in seniors over 75 years of age affected by impaired functional capacity: a Randomised Controlled Trial

GENERAL COMMENTS:

The manuscript addresses important issue of physiotherapeutic management in seniors over 75 years of age affected by impaired functional capacity. Below are the comments to each section.

Introduction

I would suggest to rewrite the introduction. The layout of the introduction is not presenting the cited references well. After each paragraph there are 6-7 references and it is not clear if they relate to the content of the last sentences or the whole paragraph. The references should be inserted after the sentence to which they refer.

Lines 70-73

Please provide reference for this statement

Lines 75-76

Please provide reference for this statement

Material and Methods

This part has to be rewritten as the same information is repeated in many places.

Participants

What were the inclusion and exclusion criteria? Please provide the description in this section.

In the title the group is described as “affected by the impaired functional capacity”. Please provide in this section how it was defined.

Lines 100-101

It is not clear how physiotherapists were blinded? In the intervention studies it is not possible to blind the physiotherapist who provide the intervention. Or the attending physiotherapist was not the one who provided the intervention?

Table 1

Please correct table 1 as there is some coloring which is not explained.

All the abbreviations used in the table should be explained below.

In the table group C seems to be intervention group as the heading is Group C OTAGO Programme and in the text it is described as control group. Please use the same name in the tables and in the text for all the groups.

The groups differed statistically in terms of the sustained falls. Could this affect the results of the intervention?

Figure 1 needs to be corrected in terms of readability.

Instruments

The part describing instruments has to be rewritten for clarity. The instruments used for the inclusion process were already mentioned in the participants part.

Please provide the psychometric properties for the instruments used.

Results

The groups differed significantly before receiving the intervention in many parameters. Why stratified randomization was not performed after the first assessment to have comparable groups?

Table 2, 3

Please correct table 1 as there is some coloring which is not explained.

Why BBS is included here as it was not measured at this measurement point.

Table 4

It has to be reorganized for clarity and checked for language.

Why BBS is included here as it was not measured at those measurement points.

Discussion

Please reorganize the discussion part for more clarity. There are many sentences which are based on the authors own experience whereas this part should discuss the results in the light of the existing literature.

Conclusions

Conclusion 3 state that the VRCRR solution can be used on an individual basis but this was not assessed in the study as the intervention was provided by the physiotherapist.

Author Response

Authors' responses to the Reviewers

The original title of the study:

Title:Physiotherapeutic management aided by VR technology vs. conventional methods, in seniors over 75 years of age affected by impaired functional capacity: a Randomised Controlled Trial

has been amended with the following wording:

Physiotherapy programmes, aided by VR solutions, applied to the seniors affected by functional capacity impairment: Randomised Controlled Trial 

Journal: International Journal of Environmental Research and Public Health 

Special Issue“Physiotherapy and Rehabilitation as Modern-Day Medical Challenge in Older Adults

             Submission ID: ijerph-1668850

The Authors have diligently addressed all the concerns raised by the Reviewers. Hopefully, the revised version of the manuscript will merit the Reviewers' satisfaction.

AUTHORS' RESPONSES TO THE REVIEWER'S RECOMMENDATIONS

REVIEWER # 3

Comments and Suggestions for Authors

REVIEW

Manuscript number ijerph- 1668850

Title: Physiotherapeutic management aided by VR technology vs. conventional methods, in seniors over 75 years of age affected by impaired functional capacity: a Randomised Controlled Trial

GENERAL COMMENTS:

The manuscript addresses important issue of physiotherapeutic management in seniors over 75 years of age affected by impaired functional capacity. Below are the comments to each section.

Introduction

I would suggest to rewrite the introduction. The layout of the introduction is not presenting the cited references well. After each paragraph there are 6-7 references and it is not clear if they relate to the content of the last sentences or the whole paragraph. The references should be inserted after the sentence to which they refer.

AUTHORS' RESPONSE:

In line with the Reviewer's recommendations, the INTRODUCTION section has been radically re-edited and supplemented accordingly.

 Lines 70-73

Please provide reference for this statement

AUTHORS' RESPONSE:

Pertinent reference has been supplemented accordingly.

Lines 75-76

Please provide reference for this statement

AUTHORS' RESPONSE:

Pertinent reference has been supplemented accordingly.

Material and Methods

This part has to be rewritten as the same information is repeated in many places.

AUTHORS' RESPONSE:

In line with the Reviewer's recommendation, the entire manuscript has diligently been re-edited, restructured, and supplemented accordingly.

Participants

What were the inclusion and exclusion criteria? Please provide the description in this section.

AUTHORS' RESPONSE:

In line with the Reviewer's recommendation, the inclusion/exclusion criteria adopted by the Authors have been moved into another section of the manuscript.  

In the title the group is described as “affected by the impaired functional capacity”. Please provide in this section how it was defined.

AUTHORS' RESPONSE:

In line with the Reviewer's recommendation, the information at issue has been supplemented accordingly, i.e. the individuals whose BBS score was ≤ 42 were pronounced as affected by functional capacity impairment.

Please consult the following study for comparison:

https://doi.org/10.1161/STROKEAHA.111.636258

Lines 100-101

It is not clear how physiotherapists were blinded? In the intervention studies it is not possible to blind the physiotherapist who provide the intervention. Or the attending physiotherapist was not the one who provided the intervention?

AUTHORS' RESPONSE:

The study was carried out by four different physiotherapists, each of them carrying out the training regimen on a different study group. The physiotherapists were randomly allocated to groups by the Principal Investigator, whereas each one of them was aware only of the specific of the therapy to be carried out.

Table 1

Please correct table 1 as there is some coloring which is not explained.

AUTHORS' RESPONSE:

In line with the Reviewer's recommendation, all tables have now been reformatted to increase their overall clarity; pertinent numerical data have been unified to 2 decimal points throughout.

All the abbreviations used in the table should be explained below.

AUTHORS' RESPONSE:

Amended accordingly throughout the manuscript.

In the table group C seems to be intervention group as the heading is Group C OTAGO Programme and in the text it is described as control group. Please use the same name in the tables and in the text for all the groups.

AUTHORS' RESPONSE:

This undue confusion has effectively been clarified. All study groups are represented as distinctive from one another.

The groups differed statistically in terms of the sustained falls. Could this affect the results of the intervention?

AUTHORS' RESPONSE:

The Authors concluded that the statistically significant difference in the frequency of falls between respective groups did not affect the actual outcome of the intervention, owing to the BBS score in the groups characterised by the highest number of falls, i.e. the CVR and OCULUS groups. In fact, those groups scored even marginally better on the BBS than the ones with fewer number of falls, i.e. the Classic and VR groups.

Figure 1 needs to be corrected in terms of readability.

AUTHORS' RESPONSE:

Figure 1. has been reformatted accordingly.

Instruments

The part describing instruments has to be rewritten for clarity. The instruments used for the inclusion process were already mentioned in the participants part.

AUTHORS' RESPONSE:

In line with the Reviewer's recommendation, this needless repetition has been deleted accordingly.

Please provide the psychometric properties for the instruments used. 

AUTHORS' RESPONSE:

It is unclear what the Reviewer has means by the psychometric properties for the instruments used.All pertinent technical data regarding the VR devices at issue are routinely comprised in the accompanying user manuals.

Besides, the actual technical specifications of the VR devices have no bearing on the study protocol whatsoever. Nor indeed have the devices been customised in any way by the investigators, either, as they opted for making use of the readily available, commercial products.

The Authors are somehow surprised that the Reviewer has brought up this subject in the first place, as they simply assumed the Reviewer would have a working command of all relevant technicalities regarding the VR equipment at issue.

Results

The groups differed significantly before receiving the intervention in many parameters. Why stratified randomization was not performed after the first assessment to have comparable groups?

AUTHORS' RESPONSE:

A fair point.

Having said that, a simple clarification is very much required here. The baseline characteristics of all enrolled study subjects were not readily available prior to their allocation to respective study groups, so consequently it was not feasible to take due note of all the variables. This issue is also addressed by Suresh (Suresh et al. 2011).

Suresh K. An overview of randomization techniques: An unbiased assessment of outcome in clinical research. J Hum Reprod Sci. 2011;4(1):8-11.

Table 2, 3

Please correct table 1 as there is some coloring which is not explained.

AUTHORS' RESPONSE:

Table 1 has been amended accordingly.

Why BBS is included here as it was not measured at this measurement point.

AUTHORS' RESPONSE:

BBS score has been moved to Table 1, even though BBS variable is also encountered in Table 3, as the Authors were keen on establishing the post-intervention values of this variable in the subjects.

Table 4

It has to be reorganized for clarity and checked for language.

AUTHORS' RESPONSE:

Table 4 has been amended accordingly.

Why BBS is included here as it was not measured at those measurement points.

AUTHORS' RESPONSE:

Please kindly note that the BBS score was not applied as the key variable in the study. As a matter of fact, the manuscript's title refers to individual functional capability, and not to the balance itself, even though balance makes one of its key components.

As already explained further above, the BBS variable was examined not only to check its compliance with the inclusion criteria, as the Authors were keen on establishing its post-intervention values in the study subjects.

Table 4 addresses the differences in BBS performance before and after the intervention.

Discussion

Please reorganize the discussion part for more clarity. There are many sentences which are based on the authors own experience whereas this part should discuss the results in the light of the existing literature.

AUTHORS' RESPONSE:

The Discussion section has diligently been re-edited, with a view to enhancing, as well as boosting the Authors' key message.

Conclusions

Conclusion 3 state that the VRCRR solution can be used on an individual basis but this was not assessed in the study as the intervention was provided by the physiotherapist.

AUTHORS' RESPONSE:

Regarding Conclusion No 3, the Authors merely made a reference to a clearly manifest potential their VRCRR programme boasts in terms of possible individual application in one's home environment. This is also addressed in the Discussion section.

Having said that, the Authors also plan to have this individual application potential tangibly verified in their forthcoming research project.

Submission Date

21 March 2022

Date of this review

16 Apr 2022 07:51:51

______________________________________________________________

Round 2

Reviewer 2 Report

Thank you for responding to my comments. Your answers are clear and your changes to the manuscript seem appropriate to me. However, the current revised version of this research is well written and well structured; the exposition is comprehensible, and the overview of related work is acceptable. Nevertheless, I have some final concerns with the discussion section; authors should highlight the discussion section and focus on the implications of their research after being exposed to such a virtual physiotherapy experience rather than just providing justifications for the findings and the limitations of their user study. Moreover, I think the title of the manuscript still needs improvement e.g., the commas are unnecessary.  

Author Response

Authors' responses to the Reviewers

The original title of the study:

Title:Physiotherapeutic management aided by VR technology vs. conventional methods, in seniors over 75 years of age affected by impaired functional capacity: a Randomised Controlled Trial

has been amended with the following wording:

Physiotherapy programmes aided by VR solutions applied to the seniors affected by functional capacity impairment: Randomised Controlled Trial 

Journal: International Journal of Environmental Research and Public Health 

Special Issue“Physiotherapy and Rehabilitation as Modern-Day Medical Challenge in Older Adults

             Submission ID: ijerph-1668850

The Authors have diligently addressed all the concerns raised by the Reviewers. Hopefully, the revised version of the manuscript will merit the Reviewers' satisfaction.

AUTHORS' RESPONSES TO THE REVIEWER'S RECOMMENDATIONS

REVIEWER # 2

Comments and Suggestions for Authors

Thank you for responding to my comments. Your answers are clear and your changes to the manuscript seem appropriate to me. However, the current revised version of this research is well written and well structured; the exposition is comprehensible, and the overview of related work is acceptable. Nevertheless, I have some final concerns with the discussion section; authors should highlight the discussion section and focus on the implications of their research after being exposed to such a virtual physiotherapy experience rather than just providing justifications for the findings and the limitations of their user study. Moreover, I think the title of the manuscript still needs improvement e.g., the commas are unnecessary.  

AUTHORS' RESPONSE:

The Authors are happy to confirm that in response to the Reviewer's suggestion they have diligently supplemented the Discussion section of the MS with the key implications of their research into VR-aided physiotherapy programme, with a view to highlighting its manifestly applied character.

The MS title has also been amended accordingly.

Hopefully, this should effectively conclude the MS revision stage, so that it could now be referred to further processing by the Editors.